# Mutational Landscape of Autism Spectrum Disorder Brain Tissue

**DOI:** 10.3390/genes13020207

**Published:** 2022-01-24

**Authors:** Marc Woodbury-Smith, Sylvia Lamoureux, Ghausia Begum, Nasna Nassir, Hosneara Akter, Darren D. O’Rielly, Proton Rahman, Richard F. Wintle, Stephen W. Scherer, Mohammed Uddin

**Affiliations:** 1Biosciences Institute, Newcastle University, Newcastle upon Tyne NE2 4HH, UK; marc.woodbury-smith@newcastle.ac.uk; 2The Centre for Applied Genomics (TCAG), The Hospital for Sick Children, Toronto, ON M5G 0A4, Canada; sylvia.lamoureux@sickkids.ca (S.L.); rwintle@sickkids.ca (R.F.W.); Stephen.Scherer@sickkids.ca (S.W.S.); 3College of Medicine, Mohammed Bin Rashid University of Medicine and Health Sciences, Dubai 505055, United Arab Emirates; Ghausia.Begum@mbru.ac.ae (G.B.); Nasna.Nassir@mbru.ac.ae (N.N.); 4Genetics and Genomics Medicine Centre, NeuroGen Healthcare, Dhaka 1205, Bangladesh; hosneara@neurogenbd.com; 5Faculty of Medicine, Memorial University, St. John’s, NL A1B 3V6, Canada; Darren.ORielly@med.mun.ca (D.D.O.); prahman@mun.ca (P.R.); 6Molecular Genetics, University of Toronto, Toronto, ON M5G 0A4, Canada; 7Genetics and Genome Biology, The Hospital for Sick Children, Toronto, ON M5G 0A4, Canada; 8Cellular Intelligence (Ci) Lab, GenomeArc Inc., Toronto, ON M5G 0A4, Canada

**Keywords:** exome sequencing, post-zygotic, somatic, germline, brain tissue, autism spectrum disorder (ASD)

## Abstract

Rare post-zygotic mutations in the brain are now known to contribute to several neurodevelopmental disorders, including autism spectrum disorder (ASD). However, due to the limited availability of brain tissue, most studies rely on estimates of mosaicism from peripheral samples. In this study, we undertook whole exome sequencing on brain tissue from 26 ASD brain donors from the Harvard Brain Tissue Resource Center (HBTRC) and ascertained the presence of post-zygotic and germline mutations categorized as pathological, including those impacting known ASD-implicated genes. Although quantification did not reveal enrichment for post-zygotic mutations compared with the controls (*n* = 15), a small number of pathogenic, potentially ASD-implicated mutations were identified, notably in *TRAK1* and *CLSTN3*. Furthermore, germline mutations were identified in the same tissue samples in several key ASD genes, including *PTEN*, *SC1A,* *CDH13*, and *CACNA1C.* The establishment of tissue resources that are available to the scientific community will facilitate the discovery of new mutations for ASD and other neurodevelopmental disorders.

## 1. Introduction

Autism spectrum disorder (ASD) is a complex neurodevelopmental disorder of early childhood onset, characterized principally by socio-communicative impairments and certain restricted behavioural patterns, but also associated with other neuropsychiatric and medical conditions. The genetic underpinning of this disorder is rapidly evolving, and the exact genetic mechanisms are complex and not fully understood, despite many ASD-implicated genes having been identified [1,2,3,4]. In brief, rare, particularly de novo, single nucleotide variants (SNV) and copy number variants (CNV) of variable size are implicated. These variants, whether SNVs or CNVs, are often variable in their penetrance and consequently almost certainly act in concert with each other and with other epigenetic and non-genetic factors. Moreover, genetic variants that are more commonly observed in the population (‘common variants’, minor allele frequency [MAF] ~1% or more) also play a role, but their discovery for ASD has lagged behind other neurodevelopmental and neuropsychiatric disorders [5]. The relative contribution of these different genetic factors and other non-genetic factors in any one individual is unclear, although cases have been described of Mendelian, or otherwise highly penetrant, single-gene/locus mutations [6,7].

Many mutations occur in the germline, inherited or arising de novo in one parent’s germ cell. However, post-zygotic mutations (PZM) may also occur during foetal development or, indeed, at any stage among those cells that undergo division as part of their life-cycle. Consequently, these events give rise to distinct cell populations, identifiable in peripheral samples, such as blood by a variant allele fraction that deviates from the 0.5 that would be expected for a typical heterozygous mutation. Using this approach, one recent study undertook ultra-deep sequencing of whole exomes from the Simons Simplex Collection and confirmed the contribution of genetic mosaicism to ASD [8]. Similarly, Lim and colleagues [9] investigated this using peripherally obtained DNA and determined that 7.5% of de novo mutations in ASD are post-zygotic, with particular genes enriched for these events.

For brain disorders, such mutations can also be directly identified in post-mortem brain tissue, a method that also allows a more detailed evaluation of their relationship to underlying histopathological abnormality and brain expression [10,11]. Such an approach has been able to identify PZMs in resected brain tissue from patients with ASD [11,12] and epilepsy [13]. In one recent study using ultra-deep, whole-genome sequencing of 59 ASD donors, the enrichment of somatic SNVs was observed in neural enhancer sequences compared to 15 control donors [14]. This adds further weight to the evidence that post-zygotic mutations (PZMs) are important in ASD’s etiology and that they may shed light on specific pathophysiological mechanisms.

In order to further investigate the role of PZMs, we used post-mortem brain samples from the Harvard Brain Tissue Resource Center (HBTRC), obtained through the Autism Tissue Program, to investigate the presence of post-zygotic mutations in ASD donors. Although there are potential challenges to extracting DNA from brain tissue, we have previously demonstrated success [15], as well as shown how postmortem tissue can yield new, previously unidentified findings [16]. In our previous study, we examined these samples for CNVs and identified that several patients had CNVs of potential aetiological significance given their predicted deleterious impact and genomic context [15]. In this current study, deep exome sequencing was performed on these same post-mortem brain specimens from donors with ASD and on normal controls to identify mutations, including those we classify as likely to be post-zygotic.

## 2. Methods

***Tissue preparation***: The postmortem tissue samples were obtained from the Harvard Brain Tissue Resource Center (HBTRC). Ethical approval was obtained from The Hospital for Sick Children, REB number 0019980189. The cohort comprised 26 samples from ASD brain donors and 15 samples from controls. Among those classified as ASD, one is reported ‘suspected’ rather than confirmed and another is diagnostically labelled Angelman Syndrome. We previously examined 52 of these samples for the presence of CNVs [15]; to facilitate comparison, identical subject labels were used. The methods of DNA extraction are described in Wintle at al. [15] Briefly, the extraction of DNA from frozen samples was undertaken either manually, using the Gentra Puregene (Qiagen Inc., Germantown, MD, USA), or by using a semi-automated procedure (Gentra Autopure LS, Qiagen Inc., Germantown, MD, USA). Deep exome sequencing was performed to an average depth of 171X at the Centre for Applied Genomics (Toronto, ON, Canada) using established methods [17]. In brief, for short read sequencing, library preparation was performed from 500 ng of DNA using the TruSeq Nano DNA Library Preparation Kit, excluding the polymerase chain reaction (PCR) amplification step, followed by sequencing on an Illumina HiSeq X platform.

***Variant Detection and Validation:*** Data analysis and base calling were performed using Bcl2FASTQ or HiSeq Analysis Software version 2-2.5.55.1311 (Illumina Inc., San Diego, CA, USA). Sequenced short reads were mapped to the hg19 reference sequence using the Burrows-Wheeler Aligner [18], version 0.7.12. Single-nucleotide variations (SNVs) and indels were detected using Genome Analysis Toolkit (GATK), version 3.4-46. All the detected variants were annotated using ANNOVAR [19], which was setup with custom databases. All downstream quality control (QC) and bioinformatics analysis on the germline variants were performed using ANNOVAR annotated variants. To detect the post-zygotic variants, we used Mutect (Broad Institute, Cambridge, USA) [20]. The algorithm was run in pairs (proband-mother and proband-father) where the intersection of the two variant files was considered to be possible somatic variants in proband. We filtered out any variants that had an alternate variant ration <10% or >40%. This allowed us to detect high-level somatic mutations from our exome data.

We undertook many iterations of random validation across all variants using Sanger Sequencing and achieved a validation rate close to 100%. In this way, although all the prioritized variants were not individually validated, the quality of the data was supported.

***Bioinformatic prediction of deleteriousness***: We followed the American College of Medical Genetics (ACMG) guideline to assess variant clinical pathogenicity using ANNOVAR annotated vcf files. In brief, most of the common variants were assessed as benign. A rare variant (gnomAD frequency < 5 × 10^−4^) was assessed as clinically ‘pathogenic/likely pathogenic’ if it was listed as ‘pathogenic’ within the ClinVar database for other unrelated patients with similar phenotypes. There is scientific literature specifying the functional role of the gene in the disease physiology and multiple bioinformatics prediction tools to support the corresponding pathogenicity cutoff for the variant. A variant was assessed as a variant of uncertain significance (VOUS) if there was not much supported information available in the literature, but it was extremely rare in the control population. Deleterious mutations were defined according to three prediction tools: Polyphen2 [21], SIFT [22], and CADD [23]. Namely, all the variants with CADD ≥ 20, and categorized as ‘damaging’ by SIFT and ‘probably damaging’ by Polyphen2, were defined as ‘bioinformatically deleterious’ by their corresponding cutoff values.

***Gene set over-representation analysis***: We explored for enrichment of post-zygotic genes using protocols implemented in both g:Profiler [24] and PANTHER [25]. Both methods examine over-representation of the gene list in each of a number of different gene-sets/pathways representing Gene Ontology (GO) annotation terms (in the case of PANTHER, this also includes PANTHER annotation terms) compared to a ‘universe’ of all genes in the genome. Fisher’s exact test was used to calculate a *p* value, corrected for multiple comparisons with a Benjamini-Hochburg procedure.

## 3. Results

A total of 26 brain specimens from individuals with ASD (19 male, 7 female) and 15 from controls (12 male, 3 female) were included. Among the ASD cases, one was diagnosed with Angelman Syndrome and one with suspected ASD. The remaining 24 had all been diagnosed with ASD. All 26 cases are also reported in Wintle et al. [15], as are all 15 controls. Identical identification numbers are used across the two papers to facilitate comparison. One individual from Wintle et al. [15], who was diagnosed with epilepsy without ASD, did not have SNV data and is not included in the subsequent analyses. No further phenotype information is available on any of the cases or controls.

A total of 2766 SNVs were reported among the 26 cases (989 synonymous and 1777 non-synonymous) and 1485 among the 15 controls (504 synonymous and 981 non-synonymous). The mean number of variants per individual was 102 (synonymous: mean, SD = 36.1, 15.3; non-synonymous: mean, SD = 71.1, 21.6) for the cases and 99 (synonymous: mean, SD = 31.6, 5.5; non-synonymous: mean, SD = 65.4, 6.7) for the controls (Appendix A).

### 3.1. Post-Zygotic Mutations

We were interested in identifying PZMs in brain tissue. By way of reminder, PZMs are those that arise at any stage after the formation of the zygote and give rise to distinct cell populations. This results in an allele fraction that deviates from the 0.5 that would be expected for a heterozygous mutation. We used the established MuTect algorithm to detect the PZMs. A total of 485 variants were classified as PZM, with 286 and 199 variants in the cases and the controls, respectively (Appendix A: post-zygotic—all; post-zygotic—predicted damaging). Two hundred and seventy-one of these variants were non-synonymous (159 in the cases and 112 in the controls), of which 25 were categorized as pathogenic according to the bioinformatic prediction (CADD, SIFT, and PolyPhen), 11 in 11 different controls and 14 in 11 cases. Of these, one was deemed likely to be pathogenic according to the ACMG guidelines [26] (Table 1). No enrichment for damaging PZMs was therefore observed in the cases versus the controls (Fisher’s Exact Test, ns). None of these variants was observed in the data from the 1000 Genomes Project, and all were either not observed or observed with extremely low frequency (<5 × 10^−4^) in GnomAD.

None of the 14 genes with bioinformatically predicted damaging mutations overlapped the ‘ASD-implicated ‘genes, a curated list of nearly 1000 putative ASD genes with varying degrees of supporting evidence [2]. Moreover, no genes were observed in the list of genes with high confidence of the ASD somatic mutations identified in Lim et al. [9]. Although none of the genes was previously implicated in ASD [27], one, *TRAK1* [chr3:42,244,127 (hg19), A > G], interacts with the known schizophrenia gene *DISC1* [28,29], and one other gene, *CLSTN3* [chr12:7310,284 (hg19), G > A]*,* was deemed likely to be pathogenic according to the ACMG guidelines. This gene encodes a calsyntenin, proteins which as a group are known to function as synaptogenic adhesion molecules in concert with others such as the neurexins, themselves implicated in ASD and other neurodevelopmental disorders [30]. Taken together with the predicted damaging nature of this mutation, it is quite possible that this was aetiologically related to the ASD diagnosed in this individual. Unfortunately, no additional clinical information is available from this subject.

We next examined those genes impacted by predicted damaging variants among cases for over-representation of different curated gene sets, as implemented in PANTHER and g:profiler. No over-representation in any of the examined gene sets was observed. We also examined protein-protein interaction pathways in STRING, but no pathway enrichment was observed. We also compared the case and control genes for any enrichment in brain expression at particular developmental time points (prenatal, early childhood, and adult in the cases versus the controls), but no differences were observed (Appendix A).

### 3.2. Germline Mutations

A total of 3749 germline mutations were identified, with 2460 in the cases and 1288 in the controls. Among those with ASD, there were 1618 non-synonymous variants, among which 200 were bioinformatically predicted as deleterious, including seven stop-gain mutations (Appendix A: germline—all; germline—predicted damaging). None of these variants was observed in the 1000 Genomes dataset, and all were either not observed or occurred with extremely low frequency (<5 × 10^−4^) in GnomAD. Thirteen germline mutations were deemed pathogenic or clinically pathogenic according to the ACMG guidelines (Table 1).

Seventy of the genes impacted by the 200 bioinformatically predicted damaging mutations are reported in Lim et al. (2017). [9] Eight of the genes impacted by these 200 variants are in the ASD-implicated list of genes (Table 1). Among these are one stop-gain mutation in *PTEN* [chr10:89,685,300 (hg19), C > A] and a missense mutation in *SCN1A* [chr2:166,856,252 (hg19), G > A]. Both *PTEN* [31] and *SCN1A* [32,33] are established neurodevelopmental genes, with mutations recorded in association with several disorders, including ID, epilepsy, and ASD. In view of their established implication in ASD and other neurodevelopmental disorders, we hypothesize that these mutations played an etiological role in the ASD diagnoses in both cases.

We also identified variants impacting *CDH13* [chr16:83,711,952 (hg19), C > T] and *CACNA1C* [chr12:2717,736 (hg19), G > A]. *CDH13* is a Cadherin gene, a group of genes which encode a class of calcium-dependent transmembrane protein; variants in this gene have been implicated in neurodevelopmental phenotypes, including ASD and ADHD [34]. *CACNA1C* is another calcium channel protein. Common variants in this gene have been identified in association with several different neuropsychiatric phenotypes [35,36], including ASD [37]. It is quite possible, therefore, that these mutations also played an etiological role in the ASD diagnosis in these cases.

Five samples were identified as having both a rare, predicted damaging single nucleotide variant and one or more pathogenic CNVs, raising the possibility of two potentially causative genetic mutations. For example, one sample had a likely pathogenic germline variant impacting *GFM1* and a CNV deletion at the *MACROD2* gene. Another sample had both a predicted pathogenic SNV in *GRM1* and a CNV loss impacting *PARK2*.

We studied the over-representation of the genes impacted by the predicted damaging variants in the cases in the curated gene sets, as described above. No over-representation was observed for either the PZM or the germline genes in any of the gene sets studied. Additionally, no enrichment in the cases versus the controls was observed for genes with expression across the prenatal, early childhood, or adult developmental stages (Appendix A).

## 4. Discussion

Brain specimens are a rare resource in neurodevelopmental disorders but can uniquely provide insight into the genetic mechanisms of disease through the ability to directly study the very tissues that underlie the manifestations of these disorders [9,10,11,12,13,15]. This includes the ability to identify the tissue of specific post-zygotic mutations. Such mutations have for a long time been recognized as an important etiological factor in focal cortical dysplasias and epilepsy syndromes [13]. However, more recently, their role in other neurodevelopmental disorders, including ASD, has emerged [11,12,15]. Much of the evidence is based on the detection of alternate allele fraction (AAF) from the deep sequenced coverage of peripherally derived specimens. In the current study, we examined donated brain tissue from patients with ASD and identified PZMs in several genes that may be aetiologically implicated in their ASD diagnoses. Although none of the genes habouring predicted-damaging PZMs was previously implicated in ASD, intellectual disability (ID), or other neurodevelopmental disorders, two, *TRAK1* and *CLSTN3*, are widely expressed in the brain.

*TRAK1* is closely associated with *DISC1*, itself implicated in schizophrenia and neural development [29,38]. Mutations in *DISC1* have been well documented in association with schizophrenia but not ASD. Additionally, one predicted pathogenic mutation in *TRAK1* has also been described in the literature in a subject with schizophrenia [39]. The implications, therefore, for neurodevelopmental disorders such as ASD are unclear, although the genetic relationship between ASD and schizophrenia is now well-established [40]. Stronger evidence for a possible role in ASD phenotypes exists for *CLSTN3,* a calsyntenin, which as a group function as synaptogenic adhesion molecules [30,41]. Specifically, *CLSTN3* is thought to be important for the normal development and function of the GABAergic and glutaminergic synapses, and although not specifically identified as an ASD-implicated gene, a de novo predicted damaging mutation in this gene has previously been described in an ASD proband [42]. Rodent models of *CLSTN3* have identified the impact of deletion on impairing cognitive function [43], offering a possible pathophysiological mechanism.

In contrast to our search among the PZMs, we did identify several germline mutations that are likely implicated in the ASD diagnoses of these subjects. For example, a mutation in *PTEN* was identified in one subject. *PTEN* is an overgrowth gene, with a variable phenotype that may include macrocephaly, tissue proliferation, including tumors, and neurodevelopmental disorders [31]. Indeed, research suggests that as many as 20% of individuals with autism who also are macrocephalic may have a *PTEN* mutation [44]. In another subject, a predicted damaging mutation in *SCN1A* was identified. *SCN1A* encodes the α-1 subunit of the NaV1.1 sodium channel and is strongly associated with epilepsy as one of the most important channelopathies [32,33]. Its most notable association is Dravet Syndrome, but other epilepsy and movement disorders have also been described, as has ASD, although the exact prevalence is unclear, and may be in the region of 24–40% [45]. We do not know if individual AN16115 had epilepsy during their lifetime.

We also identified variants impacting *CDH13* and *CACNA1C*. *CDH13* is a Cadherin gene, a group of genes which encode a class of calcium-dependent transmembrane protein. Their embryonic expression earmarks the critical role they play in axonal growth and synapse development. Although there are many members of this protein superfamily, a small number have been described in association with neurodevelopmental phenotypes, including ASD and ADHD [34]. *CACNA1C* is another calcium channel protein, encoding the α-1c subunit of the L-type voltage-gated calcium channel. Common variants in this gene have been identified in association with several different neuropsychiatric phenotypes [35,36], including ASD [37]. Moreover, mouse models have suggested a role for this gene in potentially related cognitive endophenotypes that include fear conditioning [46]. It is quite possible, therefore, that these mutations also played an etiological role in the ASD diagnosis in these cases and further reinforce the importance of cadherins and a channelopathy-mediated pathogenesis in ASD.

This current study was conducted to show the potential utility of studying patterns of mutation in brain tissue and specifically to identify post-zygotic mutations that may be of aetiological relevance. Although we failed to identify predicted damaging mutations in any known ASD or other neurodevelopmental genes, our study has successfully identified post-zygotic mutations, including a small number in genes that warrant further investigation. Our results may alternatively suggest a less widespread role for PZMs than previously predicted. However, considering the emerging evidence from epilepsy and ASD discussed above, this seems unlikely. The small sample size of the current study will have impacted on the power of this study to identify these variants. It is also possible that PZMs are of aetiological significance only for particular phenotypes, such as severe ASD, nonverbal ASD, or ASD with co-morbidities such as epilepsy. In the absence of phenotype information, we are unable to further comment on this; however, moving forward, any brain tissue resource will need to also include detailed phenotype information to facilitate the unravelling of these brain-behaviour connections. Moreover, we are not able to indicate whether undissected brain regions in these specimens are also negative for PZMs. As discussed in Wintle et al. [15]), the BA17 visual cortex was used for this current study, in part due to its easy dissection and availability. Ideally, the search for PZMs will be predicated on the identification of histopathological abnormality, which will then motivate the decision on which regions of the brain to dissect.

We hope that our results will additionally provide a resource of genetic results on postmortem tissue in ASD patients. The nature of these mutations, including their mechanism, the genes impacted, and the potential effects on brain development will only be truly resolved with greater emphasis on studies such as this one that are also able to evaluate histology and cellular function in the tissues impacted. There are several ways in which studies such as this can make further progress. The establishment of tissue resources that are available to the scientific community are making samples available, allowing increased sample sizes, and these are often from patients who have detailed phenotype information. In considering sample size, there are a number of operational complexities in establishing a tissue resource. Samples may have to be collected from different geographic locations, and these may have been stored under different conditions. Detailed information will be needed on factors that may impact the results and their interpretation. Moreover, whereas on the one hand the tissue may comprise samples resected during surgery, for other families the tissue may have been donated after the death of a loved one. This requires a sensitive and ethical process to be in place. Consequently, we anticipate that sample sizes will remain modest in comparison to other methodological approaches. This notwithstanding, the strength of this approach is the ability to directly identify PZMs in brain tissue as well as more closely study the impact of mutations in the tissue itself. In the future, it will be important to more carefully examine genetic architecture across different regions; this will be facilitated by using single cell genomic approaches. Additionally, it will be important to stratify samples by sex and developmental age, as well as to compare the genetic architecture between histologically normal and abnormal tissues.

## Figures and Tables

**Table 1 genes-13-00207-t001:** Characteristics of rare, predicted damaging mutations clinically classified as pathogenic or likely pathogenic, identified in brain tissue.

Sample	SNV								CNV			
	Type	LocusID	Ref	Alt	GnomAD ^1^	Gene	Classification	Band	Pos	Size	Type	Genes
AN00090	-	-	-	-	-	-	-	15q11.1-15q13.1	21,192,955–26,500,067	5307.10	Loss	Many
AN01093	-	-	-	-	-	-	-	7q34	142,538,076–142,561,946	23.9	Loss	*PIP*
AN01570	Germline	chr7:1510817:1510817	C	T	8.4 × 10^−6^	*INTS1*	Pathogenic	-	-	-	-	-
	Germline	chr10:89685300:89685300	C	A	NO	*PTEN*	Pathogenic					
AN03345	Germline	chr6:43008300:43008300	G	C	NO	*CUL7*	Pathogenic	-	-	-	-	-
AN03935	Germline	chr12:23687394:23687394	G	T	NO	*SOX5*	Pathogenic	15q11.1-q13.2	18,278,739–28,280,653	10,001.90	Gain	Many
AN06420	Germline	chr13:23904448:23904448	C	T	NO	*SACS*	Likely pathogenic	6q26	162,863,051–162,917,072	54	Loss	*PARK2*
	Germline	chr6:146708134:146708134	C	T	NO	*GRM1*	Pathogenic					
AN00764	Germline	chr8:133187845:133187845	G	A	NO	*KCNQ3*	Pathogenic	19p13.42	60,601,790–60,943,899	342.1	Gain	Many
AN08043	Germline	chr16:83711952:83711952	C	T	NO	*CDH13*	Pathogenic	-	-	-	-	-
AN08166	Germline	chr2:8871659:8871659	A	G	4.9 × 10^−5^	*KIDINS220*	Likely pathogenic	-	-	-	-	-
AN08873	PZM	chr12:7310284:7310284	G	A	NO	*CLSTN3*	Likely pathogenic	-	-	-		-
AN09402	Germline	chr3:158383151:158383151	G	A	8.1 × 10^−6^	*GFM1*	Likely pathogenic	15q11.1-q13.1	18,276,341–26,752,537	8476.20	Gain	Many
								20p12.1	15,760,493–15,769,465	9	Loss	*MACROD2*
AN10949	-	-	-	-	-	*-*	-	8q22.1	95,265,603–95,304,988	39.4	Gain	*CDH17*
AN13872	-	-	-	-	-	*-*	-	16p13.2	6,992,775–7,021,963	29.2	Loss	*A2BP1*
AN14613	-	-	-	-	-	*-*	-	3p14.2	60,464,015–60,502,990	39	Loss	*FHIT*
AN14762	-	-	-	-	-	*-*	-	2p16.3	51,075,080–51,087,539	12.5	Loss	*NRXN1*
AN14829	Germline	chr19:42857121:42857121	C	T	8.2 × 10^−6^	*MEGF8*	Likely pathogenic	Xp22.33	2,281,299–2,493,943	212.6	Gain	*ZBED1, DHRSX*
								15q11.1-q13.2	18,276,341–28,289,587	10,013.20	Gain	Many
AN16115	Germline	chr2:166856252:166856252	G	A	NO	*SCN1A*	Pathogenic	-	-	-	-	-
AN16641	Germline	chr12:2717736:2717736	G	A	NO	*CACNA1C*	Pathogenic	-	-	-	-	-
AN17138	-	-	-	-	-	-	-	7p21.1	16,315,368–16,370,511	55.1	Loss	*hCG_1745121/ISPD*
								15q11.2	20,302,458–21,937,715	1635.30	Gain	Many
								15q11.2	21,985,041–22,943,182	958.1	Gain	Many
								15q11.2-q12	22,989,278–23,915,837	926.6	Gain	Many
								15q12-q13.1	23,925,463–26,500,067	2574.60	Gain	Many
AN19511	-	-	-	-	-	-	-	4p15.31	22,351,359–22,429,602	78.2	Loss	*GBA3*

^1^ Frequency of variant in GnomAD database.

## Data Availability

Data available on reasonable request to corresponding author.

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
