# Peer review of "Mutational Landscape of Autism Spectrum Disorder Brain Tissue"

_genes, 2022, doi:10.3390/genes13020207_

Round 1

Reviewer 1 Report

The manuscript, genes-1556693 with the title, Mutational Landscape of Autism Spectrum Disorder Brain Tissue by Woodbury-Smith M et al., identified pathogenic related genes caused by germline and post-zygotic mutations in Autism. The authors used post-mortem human brain tissues that had been diagnosed as neurodevelopmental disorders including autism spectrum disorder (ASD) as well as the control brains. I highlighted the authors' efforts to find mutations in post-mortem patient brain tissue. However, I would suggest some minor points below.

1.    I suggest clarifying how to distinguish germline and post-zygotic mutations from the patients’ samples in the result part.

2.    Are the post-zygotic mutations that the authors suggested brain specific? It should be discussed in the discussion part with other references if those are necessary.

3.    The table should be aligned for better readability.

Author Response

  1. I suggest clarifying how to distinguish germline and post-zygotic mutations from the patients’ samples in the result part.

Thank you for raising this. We agree that it would be useful to clarify for the reader what PZMs are and their detection, and so have added this text to the results section:

“By way of reminder, PZMs are those that arise at any stage after the formation of the zygote and give rise to distinct cell populations. This results in an allele fraction that deviates from the 0.5 that would be expected for an heterozygous mutation. We used the established MuTect algorithm to detect PZMs.”

  1. Are the post-zygotic mutations that the authors suggested brain specific? It should be discussed in the discussion part with other references if those are necessary.

The post-zygotic mutations are not brain specific but instead have wider expression. Those that are more strongly brain expressed are highlighted in the discussion (TRAK1 and CLSTN3), including relevant references to their molecular characteristics and known function (discussion – paragraph 2). We hope the reviewer agrees that this expands on their brain basis in an adequate manner.

  1. The table should be aligned for better readability.

Thank you. We agree, that the table could be better aligned. We have attempted to improve this, but still feel there is room for improvement, which we were unable to achieve. We will discuss this with the production department.

Reviewer 2 Report

This article investigated the presence of post-zygotic mutations in ASD patient brain tissues, which could help to understand specific pathophysiological mechanisms for ASD. This article will be interesting for the reader of the journal, however, there are several minor revisions that the author could address as follows. 

The title of this manuscript may apply a broader meaning than your actual study. If you meant that, please leave it. Otherwise, you could specify or use some specific keywords (e.g. PZM) to help the readers better understand what is the most important point in your study. 

As the author mentioned, the sample size could be increased for future studies. Is there any specific sample number you will suggest for the future study direction? additional reference/discussion could be helpful.

Is there any difference between male and female, or between adult and young age?

Are there any references showing PZM in ASD-derived brain tissues, or your manuscript will be the first report?

Author Response

We thank reviewer 2 for their thoughtful comments on our manuscript. Our responses are in bold below:

1. The title of this manuscript may apply a broader meaning than your actual study. If you meant that, please leave it. Otherwise, you could specify or use some specific keywords (e.g. PZM) to help the readers better understand what is the most important point in your study. 

Thank you for this comment. Whilst on the one hand we were principally interested in PZMs, our manuscript does go beyond this to report on germline mutations and we hope our study will serve as both a resource of genetic results and provide impetus for this type of research in the future. We do think, therefore, that the current title captures this and hope that the reviewer agrees with our decision to leave the title unchanged.

2. As the author mentioned, the sample size could be increased for future studies. Is there any specific sample number you will suggest for the future study direction? additional reference/discussion could be helpful.

This is a very good question, and quite difficult to answer. One of the challenges with this type of research is access to tissue, and so the scale will be quite small in comparison to other approaches. That being said, resources are being established, such as the SFARI’s Autism BrainNet, that could provide opportunities to carry out a study such as this one on a larger scale. Of course, the particular importance of the study that we wished to emphasize in the discussion was the ability to both identify PZM in brain tissue itself but also more closely study the impact of mutations in the tissue.  To highlight the point you have made, we have added the following sentence to the discussion (tracked in the resubmitted manuscript):

“In considering sample size, there are a number of operational complexities in establishing a tissue resource. Samples may have to be collected from different geographic locations, and these may have been stored under different conditions. Detailed information will be needed on factors that may impact the results and their interpretation. Moreover, whereas on the one hand tissue may comprise samples resected during surgery, for other families the tissue may have been donated after the death of a loved one. This requires a sensitive and ethical process to be in place. Consequently, we anticipate that sample sizes will remain modest in compassion to other methodological approaches. This notwithstanding, the strength of this approach is the ability to directly identify PZMs in brain tissue as well as more closely study the impact of mutations in the tissue itself.”

3. Is there any difference between male and female, or between adult and young age?

Our sample size is probably too small to reflect on comparisons such as these, but with the accumulation of evidence patterns may emerge in the future. We have amended the line towards the end of the discussion to add the male female question:

“Additionally, it will be important to stratify samples by sex and developmental age, as well as compare the genetic architecture between histologically normal and abnormal tissues. “

4. Are there any references showing PZM in ASD-derived brain tissues, or your manuscript will be the first report?

We refer the reviewer to the two references cited in our introduction (reference 11).